# PIP4K2B Protein Regulation by NSD1 in HPV-Negative Head and Neck Squamous Cell Carcinoma

**DOI:** 10.3390/cancers16061180

**Published:** 2024-03-17

**Authors:** Iuliia Topchu, Igor Bychkov, Ekaterina Roshchina, Petr Makhov, Yanis Boumber

**Affiliations:** 1Robert H. Lurie Comprehensive Cancer Center, Feinberg School of Medicine, Division of Hematology/Oncology, Northwestern University, Chicago, IL 60611, USA; iuliia.topchu@northwestern.edu (I.T.); eroshchina@capefearvalley.com (E.R.); 2Institute of Fundamental Medicine and Biology, Kazan Federal University, 420008 Kazan, Russia; 3Cancer Signaling and Microenvironment Program, Fox Chase Cancer Center, Philadelphia, PA 19111, USA; igor.bychkov@fccc.edu (I.B.); petr.makhov@fccc.edu (P.M.); 4O’Neil Comprehensive Cancer Center at University of Alabama at Birmingham, Department of Medicine, Section of Hematology/Oncology, Heersink School of Medicine, WTI, Room 510D, 1824 6th Ave S, Birmingham, AL 35233, USA

**Keywords:** NSD1, head and neck cancer squamous cell carcinoma (HNSCC), PIP4K, PIP4K2B, mTORC1

## Abstract

**Simple Summary:**

This study identified the PIP4K2B protein as a key target of NSD1 in head and neck cancer. We found that PIP4K2B positively regulates the mTOR signaling pathway and promotes head and neck cancer cell growth. This study reveals a direct role of NSD1 in controlling *PIP4K2B* gene mRNA expression levels. We think that NSD1 controls the amount of *PIP4K2B* mRNA by regulating the amount of H3K36me^2^ histone methylation at the *PIP4K2B* gene. We also found that the biologic impact of PIP4K2B levels is higher in laryngeal cancer cells compared to tongue/hypopharynx cancer cells. This means that PIP4K2B plays different roles in head and neck cancer cells depending on their site of origin. Overall, the findings suggest that PIP4K2B is a novel NSD1-dependent protein in HNSCC. NSD1 and PIP4K2B have potential as targets for laryngeal cancer therapy and future cancer drug development.

**Abstract:**

Head and neck squamous cell carcinoma (HNSCC) ranks among the most prevalent global cancers. Despite advancements in treatments, the five-year survival rate remains at approximately 66%. The histone methyltransferase NSD1, known for its role in catalyzing histone H3 lysine 36 di-methylation (H3K36me^2^), emerges as a potential oncogenic factor in HNSCC. Our study, employing Reverse Phase Protein Array (RPPA) analysis and subsequent validation, reveals that PIP4K2B is a key downstream target of NSD1. Notably, PIP4K2B depletion in HNSCC induces downregulation of the mTOR pathway, resulting in diminished cell growth in vitro. Our investigation highlights a direct, positive regulatory role of NSD1 on PIP4K2B gene transcription through an H3K36me2-dependent mechanism. Importantly, the impact of PIP4K2B appears to be context-dependent, with overexpression rescuing cell growth in laryngeal HNSCC cells but not in tongue/hypopharynx cells. In conclusion, our findings implicate PIP4K2B as a novel NSD1-dependent protein in HNSCC, suggesting its potential significance for laryngeal cancer cell survival. This insight contributes to our understanding of the molecular landscape in HNSCC and establishes PIP4KB as a promising target for drug development.

## 1. Introduction

The nuclear receptor-binding SET domain protein 1 (NSD1) is a histone methyltransferase belonging to the NSD family proteins, catalyzing the mono- and dimethylation of H3K36 (H3K36me^2^) [1]. Disruptions in histone methylation, including alterations in H3K36me^2^, have been implicated in cancer development [2]. NSD1’s role in regulating oncogenic signaling pathways, such as Nuclear Factor-kappa B (NF-κB), Wnt/β-catenin, and HIF1α, as well as its involvement in Akt/mTORC1 activation and support for autophagy, has been previously established [3,4,5,6,7].

Notably, NSD1 exerts a global chromatin effect, and its depletion is associated with DNA hypomethylation in head and neck squamous cell carcinoma (HNSCC) cells [8,9,10]. Intriguingly, HNSCC patients exhibit inactivating NSD1 mutations at a frequency of 10–13%, correlating with a more favorable prognosis, particularly evident in laryngeal subtypes of HPV-negative HNSCC [8,11,12].

HNSCC, with an estimated annual incidence exceeding 900,000 cases globally [13], including over 60,000 in the United States [14], ranks as the sixth most common cancer worldwide [13]. Despite advances in therapeutic strategies, HNSCC remains a formidable and often fatal malignancy, emphasizing the critical need for a deeper understanding of its oncogenic mechanisms to drive therapeutic advances. In this study, we focused on HPV-negative HNSCC.

The NSD1-depleted phenotype has shown promise in reducing the growth of liver, breast, and esophageal cancer cells [4,5,6]. Furthermore, NSD1 depletion may enhance sensitivity to cisplatin-based chemotherapy, suggesting a potential avenue for improved outcomes in HNSCC patients [11,12]. However, despite these promising aspects, NSD1 remains a challenging therapeutic target [15,16,17], motivating our exploration of NSD1’s direct targets to uncover new options for cancer therapy.

In this study, we reveal the direct regulation of PIP4K2B by NSD1. PIP4K2B, a member of the Phosphatidyl Inositol 5-Phosphate 4 Kinases (PI5P4K/PIP4K) family proteins, plays a crucial role in PI-4,5-P2 generation, influencing phosphoinositide-signaling cascades connected to essential cellular processes such as proliferation, cell survival, membrane trafficking, and cytoskeletal organization [18]. The involvement of these family proteins in oncogenesis has been demonstrated in various cancers [18,19,20,21,22].

Given the draggability of PIP4Ks [18,23,24], our study aligns with the increasing focus on developing PIP4K inhibitors. Notably, PIP4K2B’s GTP-sensing activity contributes to metabolic adaptation and tumorigenesis, impacting chromatin organization, cytoskeleton, and plasma membrane organization [25,26]. Since PIP4K2B affects PI3K and mTORC1 signaling [18,27,28,29], and as we recently showed that NSD1 depletion decreases Akt/mTORC1 signaling [7], we investigated whether mTORC1 regulation could be PIP4K2B-dependent in head and neck cancer.

Our results demonstrate that PIP4K2B significantly influences cell proliferation and the mTORC1 signaling pathway in three distinct histological subtypes of head and neck cancer cell lines. Moreover, the overexpression of PIP4K2B in NSD1-depleted cells leads to the restoration of proliferation levels and downstream targets of mTORC1, specifically observed in a laryngeal cancer cell line. We conclude that PIP4K2B is a promising target for HNSCC therapy, especially in laryngeal carcinoma subtype.

## 2. Materials and Methods

### 2.1. Cell Lines and Cell Culture

The JHU 011 cell line, a laryngeal squamous cell carcinoma, HPV-negative, was generously provided by Dr. E. Izumchenko from the University of Chicago. The FaDu cell line, originating from hypopharynx squamous cell carcinoma, and the Cal27 cell line, derived from tongue squamous cell carcinoma (both HPV-negative), and HEK293T cells were procured from the American Type Culture Collection (ATCC, Manassas, VA, USA). All of the cell lines were cultured in a RPMI-1640 medium supplemented with 10% fetal bovine serum (FBS) and 100 U/mL penicillin/streptomycin (Gibco, Gaithersburg, MD, USA) under standard conditions (37 °C, humidified incubator with 5% CO_2_).

### 2.2. Vector Construction and Virus Production

To generate stable cell lines with inducible NSD1 knockdowns, self-complementary single-stranded DNA oligos (Appendix A) were cloned and the vectors were packaged as it was already published before [7]. To generate stable cell lines with PIP4K2B overexpression (PIP4K2B OE), CDS of the *PIP4K2B* gene was cloned into the pHage-hygro vector which was obtained in our lab before [30], using Xhol and SphI sites to generate overexpression constructs. The pHage-hygro vectors were packaged into a lentivirus system with pCMV-VSV-G (Addgene, #8454, Watertown, MA, USA) and psPAX2 (Addgene, #12260). The HEK293T cell line was used for retroviral and lentiviral system amplification with TransIT-293 Transfection Reagent (Mirus Bio, Madison, WI, USA).

### 2.3. RPPA Analysis

JHU 011 and Cal27 cell lines were utilized for RPPA (consisting of 487 validated antibodies), as these were described in our publication [7]. Specifically, the cells were lysed and prepared for analysis following the previously described protocols [31,32] from the MD Anderson Proteomics Core Facility. The obtained data were visualized using the GraphPad Prism software (Version 9.5.1).

### 2.4. Western Blot

The cells were grown in a cell culture and lysed using a lysis buffer (50 mM Tris pH 7.6, 2% SDS) supplemented with Halt Protease and Phosphatase Inhibitor Cocktail (Thermo Scientific, Waltham, MA, USA). The protein concentration of lysates was determined utilizing the Pierce BCA assay (Thermo Scientific, Waltham, MA, USA). Protein lysates with 20 μg of total protein (40 μg for NSD1 detection) were loaded per gel line and separated by a 4–20% Mini-PROTEAN TGX Gel (8% gel for NSD1) (Bio-Rad, Hercules, CA, USA). The separated proteins were transferred to the PVDF membrane using a semi-dry Trans-Blot Turbo System (Bio-Rad); for NSD1 detection, the Wet Blotting System (Bio-Rad) transfer system was used (2 h at 100 V), following the blocking in 1% nonfat milk in TBST for 1 h at RT. Incubation with primary antibodies was performed overnight at 4 °C. The list of the antibodies and used dilution included the following: NSD1 (NeuroMab, #75-280;, Davis, CA, USA: 1:750), PIP4K2B (Cell Signaling Technology, #9692;, Danvers, MA, USA; 1:1000), PIP4K2A (Cell Signaling Technology, #5527; 1:1000), PIP4K2C (Thermo Fisher Scientific, #H00079837-M01A; 1:1000), Di-Methyl-Histone H3 (Lys36) (Cell Signaling Technology, #2901; 1:1000), Histone H3 (Cell Signaling Technology, #14269; 1:1000), Phospho-p70 S6 Kinase (Thr389) (Cell Signaling Technology, #9205; 1:1000), p70S6 Kinase (Cell Signaling Technology, #34475; 1:1000), Phospho-S6 Ribosomal Protein (Ser240/244) (Cell Signaling Technology, #2215; 1:1000), S6 Ribosomal Protein (Cell Signaling Technology, #2217; 1:1000), and Vinculin (Cell Signaling Technology, #13901; 1:1500). Subsequently, the membranes were exposed to secondary antibodies for 1 h at RT (Anti-rabbit IgG, HRP-linked Antibody, Cell Signaling Technology, #7074, and Anti-mouse IgG, HRP-linked Antibody, Cell Signaling Technology, #7076; 1:1500). For detection, SuperSignal West Pico Plus Solution (Thermo Scientific) was used. The signal was captured using an autoradiography CL-Xposure Film (Thermo Scientific), with subsequent films scanning, and image quantification was conducted using Image J software (Version 1.53k). The original western blot images can be found in Appendix A.

### 2.5. RNA Isolation and RT-qPCR

To isolate total RNA, the Quick-RNA Miniprep Kit (Zymo Research, Irvine, CA, USA was used, following the manufacturer’s protocol. cDNA was then synthesized using the iScript Reverse Transcription Supermix (Bio-Rad), using 1 μg of total RNA per reaction. RT-qPCR was conducted using the QuantStudio 3 Real-Time PCR System from Applied Biosystems (Waltham, MA, USA), utilizing the SYBR Green PCR master mix (Applied Biosystems). Predesigned primer sequences for the *PIP4K2B* gene were obtained at IDT (RefSeqNumber NM_003559): 5′-TGCATCGCAAGTATGACCTC-3′, 5′-AGCTTCTGCCCTTCATTGAG-3′.

### 2.6. ChIP-qPCR

The SimpleChIP Enzymatic Chromatin IP Kit (Cell Signaling Technology, #9003) was used, following the manufacturer’s protocol. First, 3 × 10^7^ cells per sample were cross-linked with 1% formaldehyde. Second, 2.5 μg of rabbit anti-H3K36me2 antibody (Abcam, ab9049, Waltham, MA, USA) per IP was utilized, with an equivalent amount of normal rabbit IgG (Cell Signaling Technology, #2729) utilized for a control of immunoprecipitation. Purified DNA from the immunoprecipitated chromatin was subjected to qPCR analysis. The primer sequences for *PIP4K2B* gene regions are provided in Appendix A. The results were calculated as fold enrichment; target gene amplification was normalized to amplification from the IgG control qPCR.

### 2.7. siRNA Transfections

The Appendix A presents the sequences of the siRNAs utilized to silence the *PIP4K2B* gene. As a control, siRNA Universal Negative Control #1 (SIC001, Sigma-Aldrich, St. Louis, MO, USA) was used. The cells were seeded onto a 6-well plate for subsequent Western blot experiments and analysis. Upon reaching 30% confluence, the cells underwent transfection with siRNAs at a final concentration of 15 nM, facilitated by the TransIT-X2^®^ Dynamic Delivery System from Mirus. Cell lysis for Western blot analysis was conducted 72 h post transfection.

### 2.8. Proliferation Assay

In 96-well cell culture plates, 500 cells per well were seeded in complete media. Following a 24 h incubation period, the cells were transfected with siRNA targeting the *PIP4K2B* gene expression. Subsequently, CellTiter-Blue^®^ assay reagent (Promega, Fitchburg, WI, USA) was added, incubation for 1.5 h was performed, with fluorescence measurements at 560/590 nm to establish the initial (0-h) time point. This procedure was then repeated at the indicated time intervals. Proliferation was determined as a relative value, with the initial time point (0 h) set as the reference point.

### 2.9. Clonogenic Assay

The cells were seeded in 12-well plates at a density of 200 cells per well. Following a 24 h incubation period, the cells were transfected with siRNA targeting the *PIP4K2B* gene expression. The cells were allowed to incubate for 10 days to promote colony growth. After the incubation period, the cells were fixed using a solution containing 10% acetic acid and 10% methanol, followed by staining with 0.5% (*w*/*v*) crystal violet. The plates were then scanned, and the colonies were quantified using Image J software (Version1.53k).

### 2.10. The Cancer Genome Atlas (TCGA) Analysis

The Cancer Genome Atlas (TCGA) analysis was performed using UALCAN tool (http://ualcan.path.uab.edu/, accessed on 9 March 2024). To build Kaplan–Meier plots, we used the Kaplan–Meier Plotter tool (https://kmplot.com/analysis/, accessed on 9 March 2024).

### 2.11. Statistical Methods

Statistical differences between the two groups were calculated using a Mann–Whitney test. Statistical differences between the two groups at different time points were calculated by ANOVA with Šidák multiple comparison post-test. Measurement data were expressed as mean ± standard deviation (SEM). For calculations and vitalization, the GraphPad Prism software (v.9.5.1) was used.

## 3. Results

### 3.1. NSD1 Regulates PIK4K2B Protein, but Not Other PIP4K Family Members Expression in Head and Neck Cancer Cell Lines

To discover the novel targets regulated by NSD1 in HNSCC, we conducted a series of experiments focused on assessing the impact of NSD1 depletion on protein expression by using reverse-phase protein analysis (RPPA) for JHU 11 and Cal27 cell lines [7]. This analysis nominated PIP4K2B as a candidate NSD1-regluated protein (Figure 1A). Utilizing the same validated shRNA targeting NSD1, we achieved effective knockdown, as confirmed by Western blot analysis (Figure 1B,C) using three histologically different cell lines for RPPA validation: JHU 011, laryngeal, Cal27, tongue and FaDu, hypopharyngeal squamous cell carcinoma cell lines.

The PIP4K2B protein belongs to the PI5P4K/PIP4K family, which consists of two more isoforms—PIP4K2A and PIP4K2C [18]. Upon NSD1 depletion, we observed a significant reduction in PIP4K2B protein level (Figure 1B,C), while no significant change was found in PIP4K2A and PIP4K2C protein levels (Figure 1D,E). These results establish NSD1 as a regulator of PIP4K2B, revealing a novel molecular link between NSD1 and PIP4K2B in the context of head and neck squamous cell carcinoma (HNSCC).

#### 3.1.1. Mechanistic Insights into NSD1-Mediated Regulation of PIP4K2B: Transcriptional Regulation of PIP4K2B by NSD1

To unravel the precise mechanism by which NSD1 orchestrates the regulation of PIP4K2B, we initiated a comprehensive examination, focusing on the transcriptional level. RT-qPCR analysis across all three cell lines with NSD1 knockdown revealed a significant decrease in *PIP4K2B* mRNA levels, indicating a direct impact on transcription (Figure 2A).

#### 3.1.2. Epigenetic Insight: H3K36me^2^-Mediated Transcriptional Control

Given NSD1’s identity as a histone methyltransferase, which is known to regulate H3K36me^2^ levels in HNSCC [7], we conducted a ChIP-qPCR analysis using a high-quality H3K36me^2^ chip-grade antibody. Employing a set of four primer pairs covering crucial sites within the *PIP4K2B* gene (Figure 2B), our findings show that H3K36me^2^ enrichment was notably concentrated within a region spanning 1500 and 1000 base pairs upstream of the +1 transcription start site in pLKO-transfected control cells. Intriguingly, this enrichment significantly diminished in NSD1-depleted cells (NSD1 sh1), signifying a direct correlation between NSD1 presence and H3K36me^2^ deposition at this regulatory region in both JHU 011 and Cal27 HNSCC cell lines (Figure 2C,D). A compelling observation was the more pronounced reduction in H3K36me^2^ enrichment in the JHU 011 cell line compared to Cal27, suggesting potential contextual variations in the regulatory dynamics. Additionally, a subtle decrease in H3K36me^2^ enrichment was identified at the exon 9 region (+29,500) of the *PIP4K2B* gene in NSD1 knockdown cells, implying a broader impact across the gene body (Figure 2C,D).

Collectively, our results substantiate that NSD1 exerts direct regulatory control over PIP4K2B expression. This regulation occurs at the transcriptional level, with NSD1-mediated deposition of the H3K36me^2^ mark acting as a pivotal mechanism.

### 3.2. Depletion of PIP4K2B Results in Reduced Cell Growth in Head and Neck Cancer Cells

Given that PIP4Ks are potential targets for cancer treatment [18], we investigated whether PIP4K2B depletion affects proliferation in HNSCC cell lines. Using two different siRNAs for PIP4K2B and a random siRNA as a negative control (NC), we downregulated PIP4K2B in JHU 011, Cal27, and FaDu cell lines. The protein level of PIP4K2B significantly decreased (Figure 3A,B). The proliferative capacity of all three cell lines decreased over a 7-day period following PIP4K2B knockdown, as measured at the indicated time points (Figure 3C).

In a previous study, we demonstrated that NSD1 regulates cell growth and the Akt/mTORC1 pathway in HNSCC [7]. To better understand the mechanism of the PIP4K2B protein’s effect on proliferation and its potential involvement in the same signaling pathway, we evaluated well-known downstream targets of the mTORC1 complex: p70S6 kinase and S6 Ribosome protein. The Western blot results revealed that PIP4K2B depletion leads to a decrease in pT389, p70S6K, and pS240/244 S6 proteins (Figure 3D,E).

To explore additional long-term effects of PIP4K2B on cancer cell growth, we conducted clonogenic assays. JHU 011, Cal27, and FaDu cells were transfected with PIP4K2B-targeting siRNA for 10 days, fixed, and stained. Interestingly, the clonogenic assay demonstrated a significant difference between siRNA knockdown and control only in the JHU 011 cell line, while Cal27 and FaDu cells exhibited similar relative numbers of colonies (Figure 3D).

To complement our data obtained in the human cell lines in HNSCC patient tumor samples, we analyzed the TCGA HNSCC dataset. Interestingly, we found a moderate positive correlation between NSD1 and PIP4K2B (*n* = 521, r = 0.5230), NSD1 and MTOR (*n* = 521, r = 0.6174), and PIP4K2B and MTOR (*n* = 521, r = 0.5034) mRNA expression levels (Appendix A). While we have not observed the significant difference in Overall Survival (OS) and Progression-Free Survival (PFS) in patient samples with low vs. high levels of PIP4K2B mRNA (Appendix A), the analysis of the mRNA level of PIP4K2B in HNSCC tissues and normal tissues showed an increased level of PIP4K2B in tumor samples (Appendix A). The analysis based on individual cancer stage shows the same results, demonstrating the significant difference between each HNSCC cancer stage and normal tissues (Appendix A).

In summary, these data indicate that the depletion of the PIP4K2B protein results in an attenuation of cell growth in HNSCC and a concurrent downregulation of the mTORC1 signaling pathway.

### 3.3. PIP4K2B Overexpression Leads to Cell Proliferation Restoration Only in Laryngeal Cell Line

Given the observed alterations in mTORC1 signaling by NSD1 protein and previous reports by us and other groups [7,10] suggesting the potential impact of PIP4K family members on this pathway, we conducted rescue experiments to determine if the decline in proliferation and mTORC1 signaling following NSD1 knockdown is PIP4K2B-dependent.

We established inducible NSD1 knockdown and stable PIP4K2B overexpression (PIP4K2B OE) in the same three HNSCC cell lines previously utilized, using pHage as a control empty vector. Western blot validation demonstrated that the vector-encoding PIP4K2B overexpression did not interfere with the NSD1 knockdown effect, as evidenced by a decrease in H3K36me^2^ and PIP4K2B protein levels with the pHage empty vector. Similarly, the levels of NSD1 and H3K36me^2^ remained unchanged under PIP4K2B OE (Figure 4A,B).

Subsequently, we assessed the proliferation levels of each subline for every established cell line. Intriguingly, the restoration of proliferation in NSD1-depleted cells with PIP4K2B was observed exclusively in the laryngeal JHU 011 cell line. Although there was no significant difference between NSD1 sh1 and NSD1 sh1 PIP4K2B OE in sublines, PIP4K2B OE in NSD1 knockdown cells mitigated the dramatic decrease in proliferation compared to the pLKO control. Additionally, a trend toward higher cell growth was noted in the control with PIP4K2B OE in JHU 011 cells (Figure 4C). Surprisingly, Cal27 and FaDu cell lines did not exhibit proliferation restoration in NSD1 knockdown cells with PIP4K2B OE (Figure 4C). Nevertheless, a clonogenicity analysis revealed a consistent phenotype across all three cell lines, with no discernible effects on relative colony numbers after PIP4K2B OE (Appendix A).

The evaluation of changes in the mTORC1 signaling pathway after PIP4K2B OE yielded similar results. PIP4K2B OE restored the levels of pT389, p70S6, and pS240/244 S6 proteins only in the JHU 011 cell line (Figure 4D,E). These findings suggest that laryngeal cancer cells need the PIP4K2B protein, which is essential for cell proliferation especially in this cell type, by regulating the mTOR pathway.

## 4. Discussion

Recently, we have shown that many HNSCC tumors overexpress NSD1 relative to normal head and neck tissues, and that NSD1 is a major driver of cell proliferation in HNSCC [7]. In the current study, we conducted a validation of proteomics analysis (RPPA) on HNSCC cell lines, comparing those with and without NSD1. Our results reveal a robust and positive regulation of the PIP4K2B enzyme by NSD1. Furthermore, we propose that this regulation operates through the K36me^2^-dependent modulation of the PIP4K2B promoter region by NSD1.

Subsequently, we explored the functional role of PIP4K2B in HNSCC and observed a significant inhibition of cell growth upon the depletion of this enzyme across all tested cell lines. Notably, the JHU011 laryngeal cell line demonstrated a pronounced reduction in cell growth, validated through both a short-term CTB assay and a long-term clonogenic assay. Our PIP4K2B rescue experiment further highlighted that the NSD1-mediated inhibition of cell growth in the JHU011 cell line could be rescued by PIP4K2B in short-term assays, suggesting a potential dependency of laryngeal carcinoma cells on PIP4K2B for supporting cell growth.

In light of these findings, we investigated the mTOR signaling pathway, as our proteomics analysis suggested downregulation of the mTOR pathway upon NSD1 depletion, a phenomenon we confirmed experimentally [7]. Additionally, we explored the consequences of PIP4K2B depletion, drawing on prior research by Lundquist et al. that demonstrated the loss of mTOR signaling upon the deletion of PIP4K2B in normal tissues and cells in a mouse model [27].

Our investigation revealed a positive regulation of the mTOR pathway by PIP4K2B in all HNSCC cell lines tested. Intriguingly, PIP4K2B overexpression rescued mTOR downregulation specifically in the laryngeal cell line, indicating a potential heightened dependency on PIP4K2B levels in this cell type.

Moreover, we provide evidence that the PIP4K2B gene is likely directly regulated by NSD1 via a K36me^2^-dependent mechanism, aligning with prior publications from our group and others [7,33]. PIP4K2B, known for its frequent amplification and oncogenic role in breast cancer, also plays a role in lymphoma and sarcoma [34].

Given our novel findings regarding the role of PIP4K2B in maintaining the mTOR pathway and regulating HNSCC proliferation, we advocate for more in-depth studies to elucidate its role in HNSCC and other tumors.

## 5. Conclusions

In conclusion, our study contributes to advancing our understanding of the role of PIP4K2B in cancer and underscores the significance of K36me^2^ histone methyltransferases in HNSCC. Notably, a recent study by Chen and colleagues introduced a novel PIP4KA/B inhibitor, CC260, showing promising in vitro activity against p53-deficient cancer cells [35]. However, further investigations are needed to determine its efficacy in vivo, emphasizing the importance of additional studies and potential drug optimization. We propose that pre-clinical efforts to develop and test PIP4K-directed drugs, both in vitro and in vivo across various tumor types, could open novel therapeutic avenues for treating cancer and developing targeted agents.

## Figures and Tables

**Figure 1 cancers-16-01180-f001:**
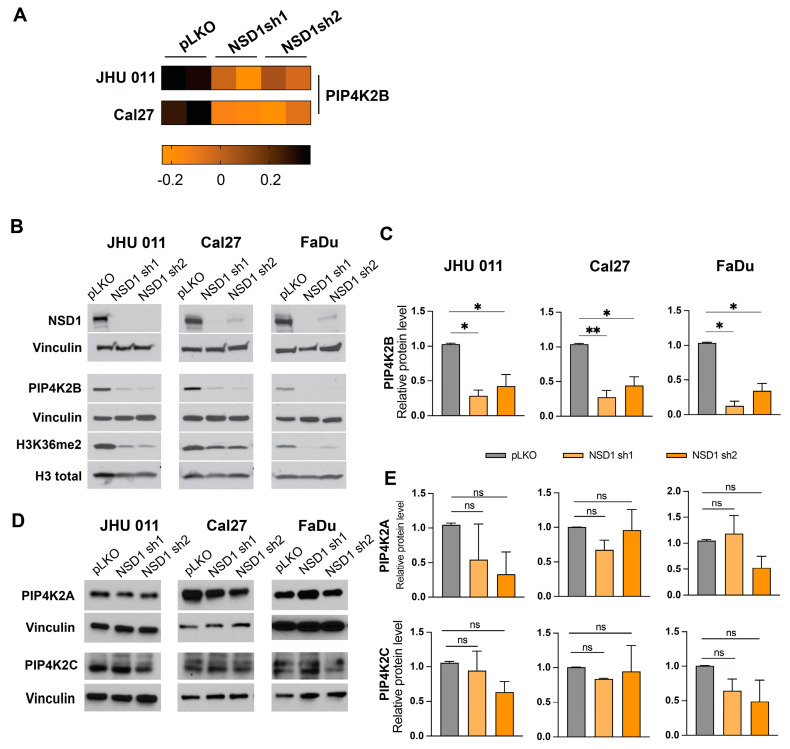
NSD1 regulates PIK4K2B protein, but not other PIP4K family members’ expression in head and neck cancer cell lines. (**A**) Heatmap of the RPPA results for PIP4K2B protein in JHU 011 and Cal27 cell lines. (**B**) Representative images of validation of the PIP4K2B protein level by Western blot in JHU 011, Cal27, and FaDu cell lines with NSD1 knockdown after 72 h doxycycline induction. (**C**) Quantification of Western blot images in (**B**). Statistical significance is determined by the Mann–Whitney test, pairwise comparing every knockdown (sh1 and sh2) with control (pLKO). (**D**) Representative Western blot images of the PIP4K2A and PIP4K2C protein levels in JHU 011, Cal27, and FaDu cell lines with NSD1 knockdown after 72h doxycycline induction. (**E**) Quantification of Western blot images in (**D**). Statistical significance is determined by the Mann–Whitney test, pairwise comparing every knockdown (sh1 and sh2) with control (pLKO). Data are performed from at least three independent biological repeats. The error bars are presented as mean ± SEM. ns—not significant, * *p* < 0.05, ** *p* < 0.01.

**Figure 2 cancers-16-01180-f002:**
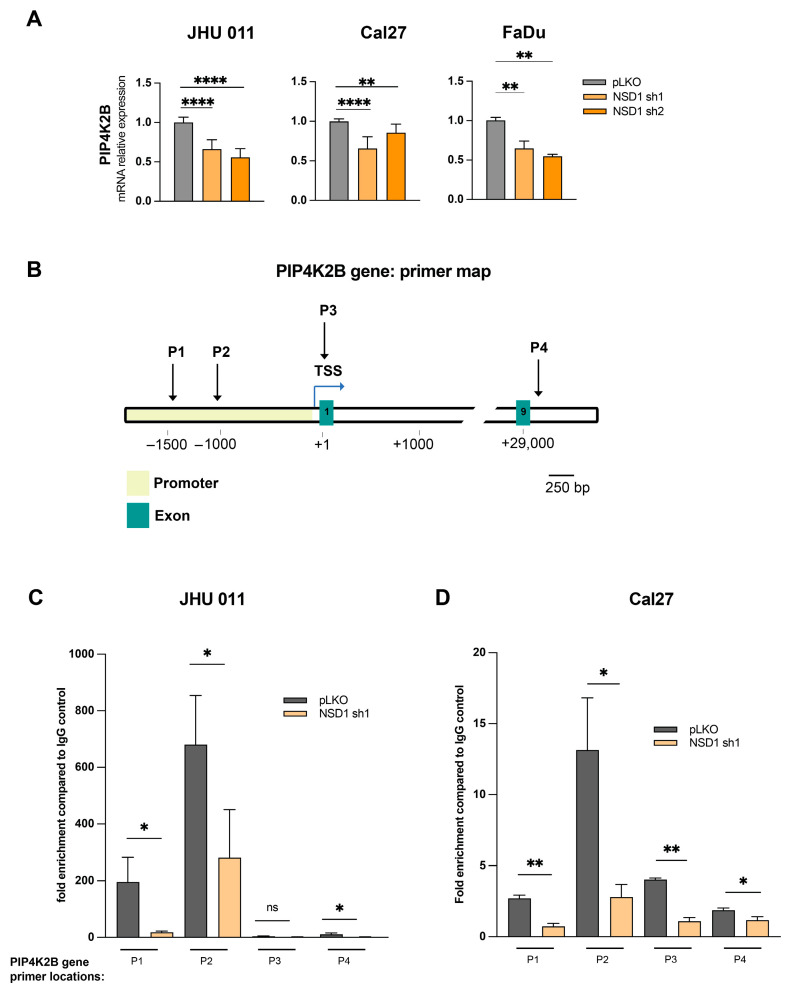
NSD1 regulates PIP4K2B in a direct manner. (**A**) mRNA level of the *PIP4K2B* gene in JHU 011, Cal27, and FaDu cell lines with NSD1 knockdown after 72 h of doxycycline induction measured by qRT-PCR. *PIP4K2B* relative level was normalized on *18S* as a control gene. Statistical significance is determined by the Mann–Whitney test, pairwise comparing every knockdown (sh1 and sh2) with control (pLKO). (**B**) Primer location map of the *PIP4K2B* gene regions for ChIP-qPCR. TSS—transcription start site; P—primers. (**C**) ChIP-qPCR with H3K36me2 antibodies on the *PIP4K2B* gene in JHU 011 cell line with NSD1 knockdown after 72 h doxycycline induction. Statistical significance is determined by the Mann–Whitney test, pairwise comparing every knockdown (sh1) with control (pLKO). (**D**) ChIP-qPCR with H3K36me2 antibodies on the *PIP4K2B* gene in Cal27 cell line with NSD1 knockdown after 72 h doxycycline induction. Statistical significance is determined by the Mann–Whitney test, pairwise comparing every knockdown (sh1) with control (pLKO). Data are performed from at least three independent biological repeats. The error bars are presented as mean ± SEM. ns—not significant, * *p* < 0.05, ** *p* < 0.01, **** *p* < 0.0001.

**Figure 3 cancers-16-01180-f003:**
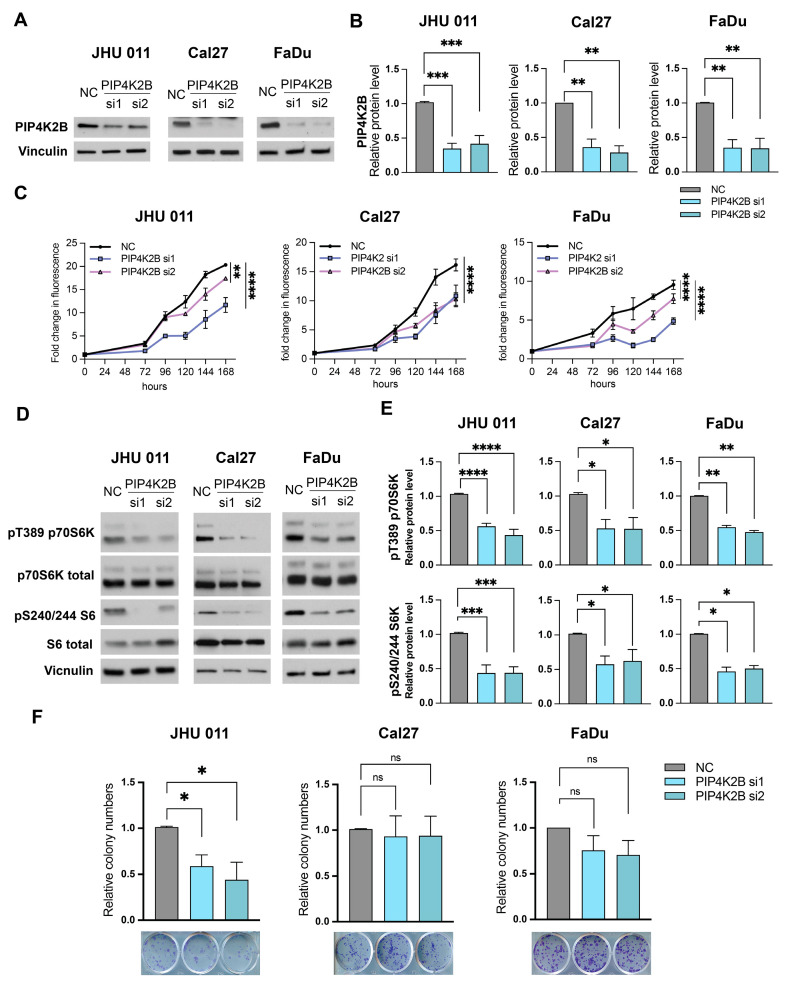
Depletion of PIP4K2B results in reduced cell growth and diminished mTOR signaling in head and neck cancer cells. (**A**) Representative images of validation of the PIP4K2B protein level by Western blot in JHU 011, Cal27, and FaDu cell lines with PIP4K2B siRNA knockdown after 72 h transfection. (**B**) Quantification of Western blot images in (**A**). Statistical significance is determined by the Mann–Whitney test, pairwise comparing every knockdown (si1 and si2) with negative control (NC). (**C**) Proliferation JHU 011, Cal27, and FaDu cell lines transfected with siRNA against PIP4K2B, as measured by CTB assay for up to 168 h, at indicated time points. Statistical significance was determined by ANOVA with Dunnett multiple comparison post-test. Each group was compared to the negative control (NC). (**D**) Representative Western blot images of p70S6K and S6 protein levels in phospho- and total isoforms in JHU 011, Cal27, FaDu cell lines with PIP4K2B siRNA knockdown after 72 h transfection. (**E**) Quantification of Western blot images in (**A**). Statistical significance is determined by the Mann–Whitney test, pairwise comparing every knockdown (si1 and si2) with negative control (NC). (**F**) Representative images of colony formation assay and quantification of the relative colony numbers in JHU 011, Cal27, FaDu cell lines with PIP4K2B siRNA knockdown. Statistical significance is determined by the Mann–Whitney test, pairwise comparing every knockdown (si1 and si2) with negative control (NC). Data are performed from at least three independent biological repeats. The error bars are presented as mean ± SEM. ns—not significant, * *p* < 0.05, ** *p* < 0.01, *** *p* < 0.001, and **** *p* < 0.0001.

**Figure 4 cancers-16-01180-f004:**
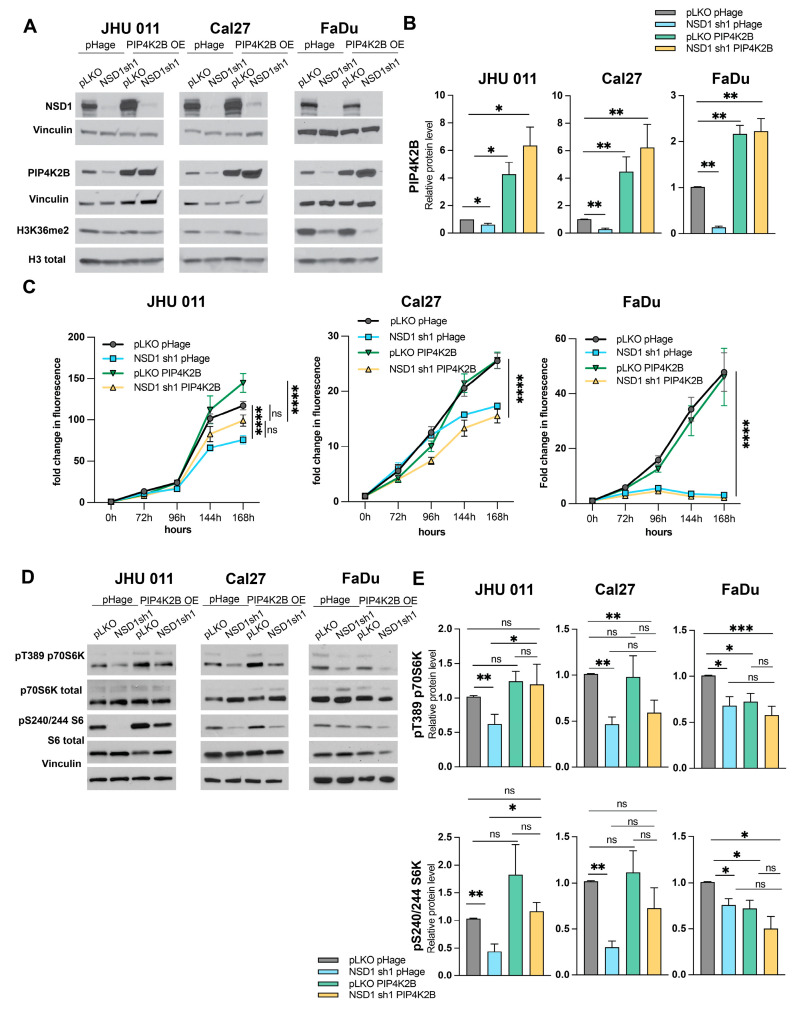
PIP4K2B overexpression leads to cell proliferation restoration only in laryngeal cell line. (**A**) Representative images of validation of the PIP4K2B protein overexpression (PIP4K2B; pHage is an empty vector) by Western blot in JHU 011, Cal27, and FaDu cell lines with NSD1 knockdown after 72 h doxycycline induction. (**B**) Quantification of Western blot images in (**A**). Statistical significance is determined by the Mann–Whitney test, pairwise comparing as it is noted on the graph. (**C**) Proliferation JHU 011, Cal27, and FaDu cell lines with PIP4K2B overexpression protein along with NSD1 knockdown, as measured by CTB assay for up to 168 h, at indicated time points. Statistical significance was determined by ANOVA with Dunnett multiple comparison post-test. Each group was compared to the other. (**D**) Representative Western blot images of p70S6K and S6 protein levels in phospho- and total isoforms in JHU 011, Cal27, FaDu cell lines with PIP4K2B overexpression and NSD1 knockdown after 72 h doxycycline induction. (**E**) Quantification of Western blot images in (**D**). Statistical significance is determined by the Mann–Whitney test, pairwise comparing every group to other. Data are performed from at least three independent biological repeats. The error bars are presented as mean ± SEM. Ns—not significant, * *p* < 0.05, ** *p* < 0.01, *** *p* < 0.001, and **** *p* < 0.0001.

## Data Availability

Data presented in this study are available from the corresponding authors upon request.

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
