# Peer review of "PIP4K2B Protein Regulation by NSD1 in HPV-Negative Head and Neck Squamous Cell Carcinoma"

_cancers, 2024, doi:10.3390/cancers16061180_

Round 1

Reviewer 1 Report

Comments and Suggestions for Authors

Head and neck squamous cell carcinoma (HNSCC) is the sixth most prevalent non-skin cancer in the world. Identification of new markers involved in the process of development and metastasis of HNSCC will significantly improve the diagnosis and therapy of patients with HNSCC.

The nuclear receptor binding SET domain family of protein methyltransferases NSD1-NSD3 is of particular interest for HNSCC, with NSD1  being amongst the most commonly mutated or amplified genes respectively in HNSCC.

Iuliia Topchu and colleagues report in their manuscript titled “PIP4K2B protein regulation by NSD1 in HPV-negative head and neck squamous cell carcinoma” some data indicating that  PIP4K2B is a novel NSD1-dependent protein in HNSCC, suggesting its potential significance for laryngeal cancer cell survival.  Furthermore, the authors show positive regulatory role of NSD1 on PIP4K2B gene transcription through an  H3K36me2-dependent mechanism.

The concept, performance and interpretation of the experiments are convincing. The used methods prove author’s expertise in molecular and cell biology to perform this study. Additionally, the description of the methods and the presentation of the experimental results are  complete and reliable. I only recommend standardizing the spelling of “Western blot”, especially in the description of figures.

Author Response

Head and neck squamous cell carcinoma (HNSCC) is the sixth most prevalent non-skin cancer in the world. Identification of new markers involved in the process of development and metastasis of HNSCC will significantly improve the diagnosis and therapy of patients with HNSCC.

The nuclear receptor binding SET domain family of protein methyltransferases NSD1-NSD3 is of particular interest for HNSCC, with NSD1 being amongst the most commonly mutated or amplified genes respectively in HNSCC.

Iuliia Topchu and colleagues report in their manuscript titled “PIP4K2B protein regulation by NSD1 in HPV-negative head and neck squamous cell carcinoma” some data indicating that PIP4K2B is a novel NSD1-dependent protein in HNSCC, suggesting its potential significance for laryngeal cancer cell survival.  Furthermore, the authors show positive regulatory role of NSD1 on PIP4K2B gene transcription through an H3K36me2-dependent mechanism.

The concept, performance and interpretation of the experiments are convincing. The used methods prove author’s expertise in molecular and cell biology to perform this study. Additionally, the description of the methods and the presentation of the experimental results are complete and reliable. I only recommend standardizing the spelling of “Western blot”, especially in the description of figures.

We thank the reviewer for this recommendation! We standardized the spelling of “Western blot” throughout the text and highlighted all corrections in green color.

Reviewer 2 Report

Comments and Suggestions for Authors

The authors found that PIP4K2B deletion decreases the viability of HNSCC cancer cell lines and replensinhg it rescuing cell growth in laryngeal HNSCC cells but not in tongue/hypopharynx cells. They claim that this is driven by mTOR axis. While we find the analysis and method thorough, we are yet not convinced about the implications of the findings and how this can help the field advance.

  1. Section “NSD1 regulates PIK4K2B protein, but not other PIP4K family members ex- 181 pression in head and neck cancer cell lines” This is a well-written section and the discovery process is thorough.

  2. It is unclear how the authors claimed that “NSD1 regulates PIP4K2B in a direct manner”.

  3. Section “Depletion of PIP4K2B results in reduced cell growth in head and neck cancer 251 cells”: We are unclear how robust the PIP4K2B dependency is.

    1. A series of genome-wide KO are performed using author’s cell lines in the dataset: DepMap (https://depmap.org/portal/). THe authors can use this to test the robustness of their findings to other HNSCC cell lines.

Author Response

The authors found that PIP4K2B deletion decreases the viability of HNSCC cancer cell lines and replensinhg it rescuing cell growth in laryngeal HNSCC cells but not in tongue/hypopharynx cells. They claim that this is driven by mTOR axis. While we find the analysis and method thorough, we are yet not convinced about the implications of the findings and how this can help the field advance.

  1. Section “NSD1 regulates PIK4K2B protein, but not other PIP4K family members ex- pression in head and neck cancer cell lines” This is a well-written section and the discovery process is thorough.

We thank the reviewer for the positive comment about our study. We would also like to comment that our novel findings of NSD1-PIP4K2B axis might be important for HNSCC cell growth lay the foundation for further studies of this pathway in cancer, and also provide a rationale to target this pathway in the future.

  1. It is unclear how the authors claimed that “NSD1 regulates PIP4K2B in a direct manner”.

Considering H3K36 dimethylation as a main function of NSD1 protein, we proposed that the decreased level of H3K36me2 at the PIP4K2B promoter gene along with NSD1 knockdown is a piece of evidence that NSD1 is a direct regulator of PIP4K2B gene. 

These data are supplemented by the data demonstrating the regulation of PIP4K2B at both the protein and RNA levels.

We agree that it would be more convincing to conduct ChIP using both NSD1 and H3K36me2 antibodies. We tried to perform NSD1-ChIP. We have several attempts with different anti-NSD1 antibodies:

  • Anti-KMT3B / NSD1 antibody, Abcam, ab70732
  • NSD1 Antibody (K47) Santa Cruz, sc-130470
  • Anti-NSD1 Antibody (N312/10), NeuroMab, 75-280.

Unfortunately, we have not been able to get a stable signal after chromatin-IP with all three antibodies despite multiple attempts and decided to focus on performing ChiP with H3K36me2 instead.

  1. Section “Depletion of PIP4K2B results in reduced cell growth in head and neck cancer  cells”: We are unclear how robust the PIP4K2B dependency is.

A) series of genome-wide KO are performed using author’s cell lines in the dataset: DepMap (https://depmap.org/portal/). THe authors can use this to test the robustness of their findings to other HNSCC cell lines.

We thank the Reviewer for the suggestion to use this tool. Unfortunately, we couldn’t find any significant probability of dependency in cell lines by PIP4K2B inside this database.

To complement our data obtained in cell lines we performed analysis of TCGA samples. We added a supplementary figure (Supplementary Figure 1) and added the descriptions of this analysis in the manuscript (Lines 280-290).

Reviewer 3 Report

Comments and Suggestions for Authors

Head and neck squamous cell carcinoma (HNSCC) is a prevalent global cancer with a low five-year survival rate of 66%. The histone methyltransferase NSD1, responsible for histone H3 lysine 36 di-methylation, has been identified as a potential oncogenic factor in HNSCC. A study using Reverse Phase Protein Array (RPPA) analysis found that PIP4K2B is a key downstream target of NSD1, which downregulates the mTOR pathway and reduces cell growth. The study also found that PIP4K2B's impact is context-dependent, with overexpression rescuing cell growth in laryngeal HNSCC cells but not in tongue/hypopharynx cells. This suggests PIP4K2B as a novel NSD1-dependent protein in HNSCC, potentially affecting cancer cell survival. The author provides more details on the data from the experiment to suggest this study and adequately addresses my concerns in this manuscript. This manuscript content is suitable for publication, and related studies have not yet been published. Some research results need further explanation in this manuscript.

1.        This manuscript mainly discusses HPV negative cell lines. If there are HPV positive cells for comparison, I believe it will be more valuable.

2.        Some errors in the manuscript content need to be corrected.

Author Response

Head and neck squamous cell carcinoma (HNSCC) is a prevalent global cancer with a low five-year survival rate of 66%. The histone methyltransferase NSD1, responsible for histone H3 lysine 36 di-methylation, has been identified as a potential oncogenic factor in HNSCC. A study using Reverse Phase Protein Array (RPPA) analysis found that PIP4K2B is a key downstream target of NSD1, which downregulates the mTOR pathway and reduces cell growth. The study also found that PIP4K2B's impact is context-dependent, with overexpression rescuing cell growth in laryngeal HNSCC cells but not in tongue/hypopharynx cells. This suggests PIP4K2B as a novel NSD1-dependent protein in HNSCC, potentially affecting cancer cell survival. The author provides more details on the data from the experiment to suggest this study and adequately addresses my concerns in this manuscript. This manuscript content is suitable for publication, and related studies have not yet been published. Some research results need further explanation in this manuscript.

  1. This manuscript mainly discusses HPV negative cell lines. If there are HPV positive cells for comparison, I believe it will be more valuable.

We thank the Reviewer for this comment. Based on the previously published data demonstrating the correlation between inactivating NSD1 mutations and a more favorable prognosis for HPV-negative HNSCC patients (Peri, 2017; Pan, 2019), in this study, we are focused on HPV-negative type of HNSCC and designed our study with only HPV-negative cells lines. We clarified this in the text (Lines 58-59).

Moreover, we believe that the role of NSD1 in HPV-positive HNSCC may be entirely different and warrants further investigation in a separate study. The study by Gameira et al. demonstrated that in HPV-positive HNSCC, cases with low expression of NSD1/2/3 are associated with inferior outcomes (Gameiro, 2021). In summary, while we agree that HPV+ HNSCC models study of NSD1-PIP4K2B is warranted, it is beyond the scope of the current study.

2. Some errors in the manuscript content need to be corrected.

We carefully checked the manuscript for any errors and corrected all potential errors.